# Cyber–Physical Distributed Intelligent Motor Fault Detection

**DOI:** 10.3390/s24155012

**Published:** 2024-08-02

**Authors:** Adnan Al-Anbuky, Saud Altaf, Alireza Gheitasi

**Affiliations:** Sensor Network and Smart Environment Research Centre (SeNSe), Auckland University of Technology, Auckland 1010, New Zealand; saud@nsu.edu.pk (S.A.); alireza.gheitasi@aut.ac.nz (A.G.)

**Keywords:** cyber–physical system, fast Fourier transform, motor fault detection, artificial neural network, distributed Internet of things

## Abstract

This research paper explores the realm of fault detection in distributed motors through the vision of the Internet of electrical drives. This paper aims at employing artificial neural networks supported by the data collected by the Internet of distributed devices. Cross-verification of results offers reliable diagnosis of industrial motor faults. The proposed methodology involves the development of a cyber–physical system architecture and mathematical modeling framework for efficient fault detection. The mathematical model is designed to capture the intricate relationships within the cyber–physical system, incorporating the dynamic interactions between distributed motors and their edge controllers. Fast Fourier transform is employed for signal processing, enabling the extraction of meaningful frequency features that serve as indicators of potential faults. The artificial neural network based fault detection system is integrated with the solution, utilizing its ability to learn complex patterns and adapt to varying motor conditions. The effectiveness of the proposed framework and model is demonstrated through experimental results. The experimental setup involves diverse fault scenarios, and the system’s performance is evaluated in terms of accuracy, sensitivity, and false positive rates.

## 1. Introduction

The concept of cyber–physical systems represents a departure from the conventional centralized control model, instead embracing a distributed architecture where intelligence is embedded directly into individual motor units [1]. This decentralization of control enables each motor to operate autonomously while simultaneously communicating and coordinating with neighboring units through a network infrastructure. By distributing intelligence across the network, cyber–physical offers several distinct advantages over traditional centralized systems. One of the primary benefits of cyber–physical is enhanced fault tolerance and robustness [2]. 

The distributed nature of cyber–physical enables superior scalability and flexibility, allowing for seamless integration of additional motors or system expansions without the need for extensive reconfiguration or infrastructure upgrades. This inherent scalability is particularly advantageous in applications where dynamic changes in operational requirements are commonplace, such as industrial manufacturing, transportation, and renewable energy systems [3]. Another key advantage of a cyber–physical system is its ability to optimize energy efficiency and resource utilization through intelligent coordination and control strategies [4,5]. By leveraging real-time data from sensors and actuators embedded within each motor unit, the cyber–physical system can dynamically adjust operating parameters [6] such as speed, torque, and power consumption to optimize overall system performance while minimizing energy wastage and operational costs [7].

Over the past thirty years, there has been significant interest in the fault diagnosis of induction motors, leading to enhancements in industrial system reliability. Numerous techniques, including those in [8,9], are suggested to determine the fault involved in electrical machines, where some solutions are commercially accessible for monitoring the behavior of induction motor behavior. Moreover, some faults are challenging to identify in their initial phases, manifesting symptoms only as induction and motor component aging accelerate.

Various machine operational characteristics can serve as indicators for monitoring the health of motors. These indicators include partial discharge, thermographic monitoring of hotspots, chemical composition, axial leakage flux, acoustic emissions, torque, power efficiency, vibration patterns, and motor current signatures [8]. Among these, motor current signature analysis (MCSA) stands out as the foremost technique [9] and is widely adopted in industry due to several advantages:Ease of implementation.Remote monitoring feasibility.Utilization of inexpensive voltage and current sensors for multiple monitoring applications.

Despite these advantages, fault diagnosis based on electric current signals is less reliable compared to other techniques like vibration monitoring. This is due to interference from noise signals, the normal operation of neighboring motors, and other sources of industrial site noise. These noises can easily disrupt a healthy signal, leading to abnormal behavior in the analysis process, potentially causing confusion. Hence, there is a strong industrial demand for robust solutions in motor fault diagnosis using electric current signal to mitigate such confusion in decision-making. Voltage and electric current are predominantly utilized parameters for indicating fault symptoms in industrial motor networks. Electric current reflects the operational characteristics of voltage in induction motors. However, the voltage at the motor terminal is influenced by both the supply voltage and the structure of the power-line network, rendering voltage alone insufficient for assessing the performance of an induction motor. Furthermore, the supplied voltage is contingent upon the motor network’s architecture and its generated voltage. 

IoT-embedded electrical drives represent a groundbreaking convergence, fusing the power of traditional electric drive systems with the intelligence and connectivity afforded by the IoT. This paper investigates the integration of the IoT, where the sensing points transmit data, and the concept of MCSA to develop a more robust and reliable solution for industrial fault diagnosis and condition monitoring.

Several investigations propose MCSA as a simpler and more effective approach for diagnosing faults in induction motors [8,9]. This technique employs a pattern recognition strategy on the electric current signal of induction motors to detect predefined fault signatures. Several recent studies have highlighted the effectiveness of MCSA in diagnosing industrial motor faults. However, a significant challenge associated with MCSA is the interference from various components within the industrial power-line network. To address this issue, certain methodologies have been suggested to enhance reliability and reduce uncertainty in decision-making. Nonetheless, interference from neighboring motors can produce frequency spectra resembling those of suspected faults. Additionally, these approaches tend to be prohibitively expensive for most low-power components of electric motors. A review survey indicates that employing a single monitoring system for many low- and medium-power electric motors is not economically viable [9].

The existing harmonic distortions found in the electric current of motors are primarily caused by asymmetries within the machinery and vibrations resulting from faults in the machine. The effectiveness of signal processing techniques relies heavily on a comprehensive understanding of both the electrical and mechanical characteristics of the machine, both when it is functioning normally and when it is faulty, across various load conditions. Several shortcomings have been identified based on the information discussed in the literature:Most research has been conducted on individual motors to diagnose their condition and performance by comparing healthy and faulty motors. There is limited research on distributed multi-motor signature analysis, where all motors are interconnected and transmit faulty signals across the network.Challenges exist in implementing diagnosis within a distributed motor network due to similarities in machine fault symptoms in the power-line network and inaccuracies in the analysis system caused by non-linear interference from industrial noise signals.Limited research has been conducted on how load variations affect the amplitudes of fault frequency components under healthy and faulty conditions.

The rest of this paper follows this structure: Section 2 offers in-depth analysis of pertinent literature reviews. Section 3 clarifies the proposed framework. Section 4 presents experimental results and discussions. Finally, Section 5 summarizes the conclusion and discusses potential future directions for research.

## 2. Related Works

Various fault diagnosis techniques are employed in real-world settings to gather useful data from targeted physical assets. Machine condition monitoring data encompass vibrations, electric currents, temperatures, pressures, or environmental data. There is a greater focus on isolated machine fault diagnosis compared to diagnosing faults in signals from multiple motors [10]. Raw sensor data undergo preprocessing before further analysis. Eliminating errors caused by background noise, human factors, and sensor faults is crucial, and appropriate features must be calculated, selected, or extracted for subsequent fault diagnosis. Once a set of features is obtained, feature selection methods are utilized to pinpoint the most effective features, streamlining the fault diagnosis process. 

To achieve accurate fault diagnosis, data must undergo a transformation into information before knowledge can be obtained. This process involves extracting or selecting fault condition indicators (features) from the acquired waveform data. Reliable features typically exhibit the following characteristics:Economical computational measurement.Interpretable in physical terms.Mathematically well defined.Insensitive to extraneous variables.Uncorrelated with other domain features.

Following the acquisition of spectrum data, various signal processing methods are employed to extract valuable feature information and analyze signal waveform data for further fault diagnosis in motors. These feature extraction techniques can generally be categorized into three groups, as illustrated in Figure 1.

Time-domain methods rely on analyzing the statistical behaviors of waveform signals over time. Key features in this analysis include root mean square (RMS) and crest factor (CF), as well as variance, kurtosis, standard deviation, and skewness. These features are derived from the distribution of signal samples over time, also known as moments or cumulate [11]. Changes in the signal can affect the probability density function (PDF) and alter the cumulate, providing valuable diagnostic information. Additional techniques for time-domain feature extraction, such as demodulation, adaptive noise cancelling, filter-based methods, and stochastic techniques, are discussed in the literature. However, one limitation of time-domain feature extraction is its difficulty in detecting faults, especially in their early stages, due to a lack of visible symptoms. This technique may be more effective when extracting short-duration features from the signal.

Frequency-domain features have the advantage of addressing the limitations of time-domain analysis. These features rely on the information provided by periodic waveform signals generated by localized faults, along with distinctive frequency characteristics. Changes in frequency-domain parameters can indicate the presence of faults, as different faults exhibit unique frequency spectrums. Early fault detection and diagnostics can be performed using these parameters. To improve spectrum analysis results, various methods such as frequency filtering, sideband structure analysis, demodulation, and descriptive representation techniques are often employed [12]. Different types of frequency spectra, including power spectra and high-order spectra, have been developed. While the traditional method of generating a power spectrum involves using the discrete Fourier transform (DFT), additional techniques like the maximum entropy technique can also be utilized. Time-frequency methods offer the capability to characterize machinery fault signatures in both time and frequency domains, particularly when the signal is non-stationary. The conventional approach involves using time and frequency distributions to represent the signal’s energy in two dimensions. 

Various fault diagnosis techniques have been utilized to address single and multiple faults in industrial machinery systems. These techniques primarily include signal-based, model-based, knowledge-based, and hybrid methods. Additional classifications of these methods are detailed in Figure 2.

Signal-based fault diagnosis methods rely heavily on signal processing techniques and typically require predefined thresholds. Signals are characterized by features, and when these features exceed their predefined boundaries, it may indicate an abnormal condition. Various methods, including vibration analysis, MCSA, axial flow analysis (AF), torque analysis, noise monitoring, and inverse sequence impedance, are available for signal analysis-based fault diagnosis [13,14]. 

In recent years, MCSA has gained attention from researchers in academia and industry as a method for motor condition indication, similar to vibration monitoring but without accessing the motor directly. Stator current is commonly measured for motor protection, and it is readily available when the motor is controlled by a drive system. Several authors have discussed fault detection in multiple motors using MCSA, but they typically isolate motors from the system. Gheitasi et al. discussed fault diagnosis using MCSA on multiple motors simultaneously, focusing on diagnosing single faults and noise levels in each motor [13]. However, they did not address uncertainty management due to the complexity of different faulty signals in their research.

Model-based fault diagnosis relies on dynamic system models and benefits from actual system and model outputs. Comparing simulation and actual data outputs allows for visualizing the condition of a motor. Dynamic models can be developed using physical modeling, system identification, and parameter estimation. However, the accuracy of the model determines the effectiveness of the diagnosis system, and modeling uncertainty arises from challenges in obtaining knowledge from monitoring processes in noisy environments.

Knowledge-based model strategies, utilizing human brain-like knowledge, are often used for machine fault diagnosis. In real-time diagnostic practices, engineers with expertise in diagnosing motor faults apply these strategies and methods, especially when dealing with dynamic signal conditions to reduce uncertainty. Various studies have explored fault diagnosis in isolated induction motors using different techniques, with artificial neural networks (ANNs) being a commonly used method due to their excellent pattern recognition abilities, especially with fuzzy and indefinite signals. The ANN’s special characteristics make it suitable for applications in information fusion and fault diagnosis.

Several studies propose ANNs as a knowledge-based approach for diagnosing single or multiple motor faults. These studies involve mapping various fault symptoms in an individual motor to make a diagnosis. Eldin [14] developed a diagnosis system using an ANN for isolated machines, training the network with RMS measurements of electric current, voltage, and speed to identify motor rotor faults. However, voltage faults are only detectable in steady-state conditions, not during dynamic loads. Another study by Arabaci focused on the impact of rotor faults on electric current in the frequency domain, utilizing an ANN in steady motor operation [10]. This research highlighted significant frequency components on the frequency spectrum associated with broken rotor bar faults as symptoms, which were then used as inputs for a supervised ANN architecture. The proposed technique effectively diagnosed rotor faults and distinguished between different fault types with reasonable accuracy.

Fault diagnosis studies using ANNs have shown success when applied to individual motors. However, when dealing with a distributed network, the complexity increases due to the non-linear manipulation of signals, potentially leading to confusion from multiple similar motor faults [15]. This makes it challenging to precisely attribute a fault to a specific machine or type. To address this issue, this research employs a distributed ANN approach to accurately identify fault types and locations within a motor network, leveraging significant motor features to overcome the complexity and mixture of signals from multiple sources.

Broken rotor bars (BRB) or end-rings can result from frequent on-line motor starts, mechanical loads, or manufacturing defects in the rotor cage [16]. While BRB may not cause immediate motor failure, loose pieces can strike the stator end winding, leading to serious damage or injury [17]. An air-gap eccentricity (ECE) fault occurs when a gap forms between the rotor and stator, leading to an imbalance that causes vibrations and noise in induction motors [18]. As eccentricity increases, the unbalanced radial forces may result in contact between the rotor and stator, causing potential damage [19].

In an industrial power-line network, a faulty wave signal shows a strong relationship between electric current and voltage waves with specific impedance characteristics [20]. The input impedance of multiple connected electric motors is an important parameter, especially near the grid frequency (50/60 Hz) [21]. Input impedance becomes more crucial at higher signal frequencies due to the widespread use of motor variable speed drives. Fast switching between power semiconductor phases injects signals with high energy content and a large frequency spectrum into motor feeder cables, potentially causing electromagnetic emissions, inverter problems, and damage to motor insulation windings [22]. 

Numerous solutions utilizing wireless sensor networks (WSNs) for industrial machinery have been developed and documented by both commercial entities and individual researchers [23]. These solutions primarily focus on data acquisition and signal transmission, with feature extraction and data fusion typically handled by a central computer. However, some approaches incorporate fault diagnosis upon sensor data acquisition, aiming to streamline data processing and conserve node power. Industrial wireless sensor network (IWSN) solutions for motor fault diagnosis and condition monitoring must account for the specific requirements of industrial processes and the unique characteristics of motors [24]. While certain industrial processes demand high sampling rates, rapid data transmission, and data reliability, an IWSN faces constraints such as limited computational capabilities, radio bandwidth, and battery energy. Balancing these high system requirements with the resource constraints of IWSNs poses a challenge.

The recent literature highlights the application of IWSNs in various areas including machine condition monitoring, fault diagnosis in pumping systems, manufacturing machines, smart grids, power plants, and structural health monitoring. For instance, one study focuses on electric current and vibration-based data acquisition to monitor rotating machinery in power plants, employing a sensor-level data fusion algorithm for condition diagnosis [25]. Another study introduces a diagnosis solution for observing rotating machinery in power plants, utilizing electric current and vibration signature data acquisition. This system monitors motor vibration and stator electric current signatures from different motors, employing node-level feature extraction techniques and a neural network classification method for training and uncertainty management [26]. Decision-level fusion is implemented at the node coordinator, with training conducted efficiently in offline mode. In this approach, fault states such as BRB and ECE need to be manually detected at experimental motors under various load levels. Overall, these studies illustrate the complexities involved in deploying an IWSN for industrial machinery fault diagnosis and condition monitoring, highlighting the need for innovative solutions to address both the system requirements and resource constraints inherent in this domain.

## 3. Distributed Motor Network Signature Analysis and Diagnosis of Fault Types

This study examines the layout structure of a typical industrial multi-motor power-line network to illustrate how fault signals propagate and can be identified along the main power bus. The network serves as a conduit for signals, influencing the behavioral signals of other motors based on their proximity. Various aspects are considered for a configuration of interconnected induction motors via a supply bus, with multiple measuring points used to observe motor behaviors. The model includes a main power bus, segmented bus bars, and connected motors. To assess observations at each measuring point and analyze diagnosis effectiveness, different numbers of measuring points are assumed at specific locations to observe individual motor behaviors within the same bus, as shown in Figure 3. 

This work focuses on analyzing motor conditions using the MCSA method instead of all available diagnosis methods. Significant feature characteristics are extracted from motor current signals using MCSA theory, which helps in identifying fault patterns. These features are valuable for understanding motor behavior within networks and are used in data fusion techniques. The analysis includes individual and combined examination of fault patterns in low- and high-power induction motor networks, highlighting the impact of faulty signals on neighboring motors and the need to filter out noise for accurate fault detection. The pattern recognition approach is extended to all sensing points in the network using a distributed signature analysis diagnosis strategy.

Making decisions at single points can often lead to serious failures in identifying fault types and pinpointing the origin of fault signals within a distributed power-line network. However, utilizing multiple measuring diagnosis points can enhance the accuracy of fault diagnosis by capturing interdependent data. This approach helps differentiate the fault’s origin and effectively isolate individual motor faults with satisfactory precision.

The focus of this study is to emphasize the potential accuracy and success in detecting the presence of fault signals by analyzing the associated spectrum at the node level. It is essential to identify possible diagnostic challenges caused by factors such as noise, the propagation of fault signals, various paths available on the power-line network, interference from neighboring nodes, and the occurrence of multiple similar faults across different nodes. Subsequently, there is a need to accurately locate faults within the power-line network. In this research, a distributed fault detection framework for multiple-motor architectures has been employed, utilizing wireless sensor network (WSN) connectivity for fault diagnosis, as illustrated in Figure 4.

Figure 4 illustrates the concept of distributed signature analysis, which involves observing electric current signals at multiple sensing points to diagnose faults more accurately. This approach utilizes the potential accuracy of direct analysis and diagnosis when multiple points are available, and it helps to clarify fault symptoms caused by noise in the network. The focus is on identifying faulty motors within the power-line network and estimating the impact of fault signals on in-network motors using signal processing. The investigation also considers signal attenuation and the propagation of fault signals in motor networks, identifying different paths for signal propagation. The framework highlights the role of WSN nodes in data collection, fault signature creation, noise level identification, fault symptom diagnosis, monitoring neighboring node behavior, and alerting the coordinator node in suspected situations. This distributed diagnosis solution is expected to provide more effective and reliable results for motor networks, especially in cases where direct measuring methods are not available.

The complexity of motor network modeling in industry makes using a fixed number of hidden layer nodes inefficient and time-consuming for training and classification. Instead, neural networks are used to efficiently recognize abnormal voltage signal representations. This section focuses on discussing neural network architectures and methodology specifically for diagnosing targeted fault types in multi-layer perceptron (MLP), rather than providing a comprehensive overview of typical neural networks. Each input consists of dynamic feature values from multiple motors, depending on their power. Multiple architectures were tested to select the optimal number of hidden layer nodes for better performance, balancing between adequate training and efficiency. The demonstrated NN architecture includes input, hidden, and output layers, with dynamic hidden layers associated with each bus in the power-line network, analyzing the condition of respective bus motors, as shown in Figure 5.

## 4. Distributed Motors Network Mathematical Modeling

The dynamic characteristics of motors within a distributed power-line network underscore the importance of implementing a time-variant methodological approach to address potential anomalies. To effectively analyze the spectrum of multiple motors at each bus, the following steps should be followed:Initially, data must be gathered from various sensing points across the power-line network. Subsequently, significant sideband frequency points and their corresponding amplitude values are identified as potential fault signals stemming from faults in multiple motors.The subsequent phase involves consolidating the significant frequency sideband values from multiple sensing points within the relevant bus associated with the targeted faults. A threshold is established for all sensing points and collected data. Any frequency sideband components not aligning with significant frequency band patterns are removed from the signal, and the remaining signal is categorized based on the reference fault pattern.The significant frequency sideband amplitude values at different sensing points are examined. If these values significantly deviate from the defined healthy range, it suggests that the fault originates from either the motor itself or a neighboring motor within the same bus. Otherwise, there is suspicion that the signal originates from other buses.The intensity and impact of faulty sidebands at each motor are contingent upon the distance between the various buses and motors. For instance, in the same bus, if a high-power motor transmits the faulty signal, similarly powered motors would exhibit comparable influence, albeit with slight differences in amplitude values. Conversely, a low-power motor would exert a greater influence, indicating a potential fault. Similarly, if a low-power motor transmits a faulty signal, a high-power motor in the same bus would have diminished influence due to its higher horsepower (HP), thereby introducing uncertainty as to whether the suspected sideband components emanate from their respective motor or from others. This ambiguity may impede the accurate identification of the source of fault indices.To pinpoint the faulty motor within the network, the suspected bus zone is isolated after assessing multiple sensing points. Subsequently, all motor signatures within the same bus are scrutinized to identify the motor exhibiting ambiguous behavior. However, owing to the uncertainty surrounding the faulty sideband, the actual speed of the suspected motor must be measured and compared with the synchronized speed based on its mechanical properties. If the speed falls within the normal range, it indicates that the sideband frequency has originated from another motor. However, the speed factor alone is insufficient to conclusively determine the motor’s fault status. Therefore, significant points are classified as indicators of potential faults, and a fault index is generated based on multiple frequency points. A graphical representation illustrating multiple sensing points across the network to ascertain the precise location of a motor is depicted in Figure 6.The diagram depicted in Figure 6 illustrates the multi-path propagation of signals and sensing points $S_{1},\;S_{2},\ldots ,S_{n}$ across the network. This diagram showcases numerous sensing points distributed throughout the network, each corresponding to a specific sub-bus. Each link within the diagram is composed of four branches, delineating various route paths $R_{1},\;R_{2},\ldots ,R_{n}$ with different lengths between buses $L_{1},\;L_{2},\ldots ,L_{n}$ and denotes the characteristic impedance at each sensing point and other relevant parameters. By understanding the configurations of sensing points and their interconnections, it becomes feasible to anticipate different combinations and evaluate the fault strength at each point, particularly when the distances between motors are known within an industrial setting. Additionally, this analysis aids in determining the direction from which the propagated signal originates. However, when the distances between motors and buses are unknown, solely sensing the values at each point may not suffice to distinguish between fault signals. In such instances, incorporating the attenuation factor becomes crucial. This factor plays a pivotal role in elucidating the locations of faulty motors within the network, a topic that will be further elaborated upon in subsequent sections.

The fault index serves as a measure of the intensity of each respective fault signal at every measurement point. It can be displayed in tabular format to depict multiple sensing points corresponding to the actual positions of various motors, as illustrated in Table 1.

Figure 7 serves as a reference for constructing a mathematical model to describe the propagation and impact of faulty signals. Within this illustration, a single motor ($M_{1.3}$) is posited as the source of a faulty signal stemming from a rotor fault, potentially leading to a voltage decrease in the system.

## 5. Experimental Multi-Motor Testbed and Results

To demonstrate the transmission of faulty signals across a power-line network and identify various fault types, researchers at the Auckland University of Technology (AUT), SeNSe laboratory, New Zealand modeled and executed an experimental motor network. They constructed a setup with two different sizes of single-phase induction motors connected to the same primary power bus. A total of nine induction motors was utilized to simulate a multi-motor network, distributed across three sub-buses, as illustrated in Figure 8.

Each sub-bus comprised three induction motors, consisting of two 15 w motors (Model S7I15GE-S12) and one 25 w motor (Model S8I25GE-S12), as shown in Table 1. The inclusion of motors with differing power ratings aimed to investigate the impact of higher-rated motors on lower-rated ones. Each motor was equipped with its own measurement point to capture data accurately. Load torque, induced through brakes, was applied to induce motor vibration. Data were sampled at a rate of 25,000 samples per second, with each measurement requiring one second for storage onto a flash drive. Faults were deliberately introduced by inducing vibration through the brakes and by misaligning the rotor.

A practical experiment was conducted as part of the experimental procedures. The Tektronix storage oscilloscope (TDS2012B) was utilized to record measurement data onto a flash drive. Simultaneously, Tektronix current probes (A622) were employed for capturing electric current data, along with manual data collection. The speed of each motor was measured using a digital laser tachometer (Standard ST-6234B; CEM: Shenzhen, China). Subsequently, all gathered measurement data were organized into distinct Excel spreadsheet files for each motor. Further details regarding the instruments used for data acquisition are presented in Table 2.

The WSN system’s development involves meeting hardware and software prerequisites. To create the application, a testbed environment is established utilizing readily accessible hardware components as shown in Figure 9. Subsequently, a software application is crafted and assessed on a hardware platform. This segment outlines the hardware system utilized, followed by the software devised and executed for motor fault diagnosis applications. The hardware setup employed consisted of an Arduino development board serving as the foundation. Positioned atop it was the wireless sensor SD board shield, followed by an XBee module on the uppermost tier. The setup utilized the Arduino development board as its foundation, topped with the wireless sensor SD board shield, and further layered with an internet-enabled XBee module. Figure 9 depicts the experimental arrangement of Arduino, incorporating a current sensor for the acquisition of current signals destined for Arduino analysis.

Utilizing the XBee Radio Frequency (RF) unit, seamless connectivity is established between the host device and Arduino unit via asynchronous serial communication. Data from speed sensors and current sensors are transmitted to the module’s universal asynchronous receiver–transmitter (UART), with speed sensor data injected through digital pin 6 and current sensor values via pin A_0_. Prior to analysis, analog signals are converted to digital form by an analog-to-digital converter. Configuration settings, including parity, baud rate, and data bits, must align between the microcontroller and XBee module for successful data transmission. In this setup, the XBee module serves as both coordinator and end-node, configured at a data rate of 9600 baud using Cross-Platform Configuration and Test Utility (X-CTU v6.2.1) software.

A setup utilizing the XBee RF unit to interface with a host device via a logic-asynchronous serial port. The Arduino unit configures and communicates with voltage-compatible UARTs, enabling connection to various serial port devices. Speed sensor signals enter the module UART via digital pin 6, while current sensor values come from pin A_0_ as analog signals. The analog-to-digital converter converts these to digital form for analysis. Data transmission is idle when no signal data are being transmitted. Configuration, including parity, baud rate, data bits, and stop bits, is crucial for compatibility between the microcontroller and XBee RF module. Figure 10 illustrates the setup with X-CTU software configuring the XBee module as both coordinator and end-node, operating at a data rate of 9600 baud.

The XBee module operates in three distinct modes: idle, transmit, and receive. In idle mode, the module remains inactive, neither transmitting nor receiving packet data. However, it can transition to other modes based on specific conditions:Transmission Mode: When sequential data are prepared for transfer into the buffer in packet form.Receiving Mode: When the coordinator receives structured RF signals via the XBee antenna.Sleeping Mode: Enabled solely on end-nodes.Command-Based Mode: When a valid mode sequence command is distributed to all end-nodes.When the node receives sequential data from sensors, it enters transmit mode. Here, the data are converted into packets, prompting the RF module to exit idle mode and enter transmission mode for data transmission. The destination address determines the receiving node. Prior to transmitting packet data, the module verifies that the route and 16-bit network address for the receiving node have been appropriately confirmed and established. Figure 11 illustrates the process flow when the node is in transmission mode.

The diagrams in Figure 12 illustrate the functions of coordinator and end-node devices within a network based on a specific set of configurations. In this study, the WSN system employs a byte-oriented data packet transmission method, which involves assigning start and end flags, an address, control byte, and information. These flags signify the beginning and end of each packet. Within this communication framework, users can transmit commands in a command frame structure from the coordinator to the end-node device to initiate various tasks, such as initiating data collection, retrieving stored readings from erasable programmable read-only memory (EPROM), or querying electric current and speed sensor values. Upon receiving such a command packet, the end-node verifies the address. If the PAN ID matches a predefined address, it proceeds with the specified operation; otherwise, it disregards the packet. Subsequently, the end-node forwards similar packets containing relevant information to the coordinator. On the computer side, the coordinator validates the sender’s authenticity by inspecting the received data packets.

The collected data are accumulated sequentially over one second into multiple samples and stored in the Arduino buffer. Once all packets are transmitted to the coordinator, the end-node enters sleep mode until further instructions are received. The system employs byte-oriented data packet transmission, utilizing start and end flags to mark packet boundaries. Data are transmitted from the end-node to the coordinator as digital packets, with each measured data packet occupying two bytes of memory at a data rate of 9600K baud. Figure 12 demonstrates how data are transmitted between different layers of the network protocol stack, ensuring efficient and reliable communication. Figure 13 illustrates the packet structure and data communication flow, highlighting the various components that make up the packet, including headers, payload, and error-checking mechanisms.

Each packet comprises sixteen data bytes, with eight bytes allocated for collected data, two for header timestamp, three for packet information, and two for cyclic redundancy check (CRC) markers to ensure data integrity.

During the initial phase of visualizing the test, data were acquired using an oscilloscope and a hand-held current probe. The electric current data for all motors were captured using two oscilloscopes and current probes, as illustrated in Figure 14. Considering limitations in the measurement apparatus, data were collected at different time intervals from distinct motors, potentially impacting the quality of sidebands associated with the electric current signal. However, this variation does not necessarily degrade the quality of the collected data, as the rate of change is expected to decrease reasonably during motor internal fault diagnosis. The electric current waveform was captured using a probe, after which the built-in fast Fourier transform (FFT) function of the oscilloscope was utilized to examine the sidebands. Figure 14 depicts real-time data captured from all targeted motors under no load, with frequency (Hz) on the horizontal axis and amplitude (dB) on the vertical axis.

As depicted in Figure 15, there is no notable sideband evident in any electric motor, given that all data were collected while the motors were operating without any load torque. As previously mentioned, the absence of load torque precludes the emergence of fault sidebands. Consequently, faults were intentionally induced in Motor 1. Figure 15 illustrates how the faulty signal from Motor 1 affects the others via the main power-line as described below:

As seen in Figure 15, Motor 1 exhibits the highest fault sideband at specific frequencies (29 Hz and 71 Hz) compared to the other motors, indicating clear BRB fault symptoms. Additionally, the influence of sideband amplitude decreases with distance from Motor 1. The distances between the nine motors have already been established.

A case study was conducted to validate the concept and illustrate how a noisy fault signal propagates and impacts neighboring motors’ characteristics based on their proximity. Motor 1 was designated as the faulty motor, and artificial faults were introduced to simulate a real-world motor fault scenario. The following steps were taken to create this scenario. For this purpose, two different sizes of motors were selected from the same bus to observe the influence of a fault signal on other motors, as shown in Figure 16. The network depicted in the following figure was chosen for the experiment.

Figure 17 illustrates the presence of mirrored sidebands around the fundamental frequency in all motors, with varying amplitudes, as a result of BRB (Motor 1) and air-gap eccentricity (Motor 3) faults introduced at full-load. These faults were analyzed to assess the attenuation strength of the fault spectrum and validate the diagnostic capability under different fault conditions. In addition to motor-level sensing points, other locations were monitored to evaluate signal strength and observe the significant frequency sidebands associated with these faults.

Figure 16 and Figure 17 illustrate the electric current spectrum and amplitude values for all motors. As shown in Figure 18, the BRB fault signal mirroring sideband consistently appears at 43 Hz and 57 Hz, but with varying amplitudes depending on the motor’s size and proximity to Motor 1. Additionally, the ECE fault, originating in Motor 3 (at 9 Hz and 91 Hz), is evident in the electrical current signals of other motors, as shown in Figure 19.

Table 3 below displays the training data used to improve the decision-making capabilities of all motors.

Figure 20 and Figure 21 illustrate the performance and confusion matrices for the training, testing, and validation phases of each motor.

In Figure 21, Each diagonal cell represents the number of cases correctly classified by the neural network, indicating whether a motor’s feature condition is healthy or faulty. The red non-diagonal cells show the number of cases misclassified by the ANN, where the feature condition was not correctly identified. The final blue cell in each matrix shows the total percentage of correctly classified cases in green and incorrectly classified cases in grey. Table 4 demonstrates a high accuracy rate (84–88%) in fault detection within the feature vector. This showcases the effectiveness of the ANN in minimizing uncertainty and enhancing decision-making for multiple-motor fault diagnosis, particularly in complex scenarios involving multiple fault frequency signals. Figure 22 validates the developed prototypes by comparing different features from each motor. The figure presents a comparison between MATLAB (R2014b) simulation and experimental results, confirming the system’s ability to diagnose the condition of all motors within a network.

The system’s ability to distinguish between different classes is compromised due to these interfering signals, resulting in a higher error rate. Additionally, the complexity and variability of the signals from different motors add to the difficulty, making it hard to maintain consistent diagnosis accuracy across all motors. Therefore, the error rate is slightly higher due to the intricate nature of the signals within the distributed network.

## 6. Conclusions and Future Recommendations

This research paper delved into fault detection in distributed motors, leveraging the Internet of electric drives (IoED) concept. It proposed a cyber–physical system architecture and a mathematical model for effective fault identification. The architecture adopted a DIoT approach, while the model employed FFT and an ANN to improve the accuracy and sensitivity of fault detection. The mathematical model captured the complex interactions between distributed motors and their edge controllers. FFT was used for signal processing, extracting frequency features that indicated potential faults. Simultaneously, an ANN-based fault detection system was incorporated, utilizing its pattern recognition and adaptability capabilities. The experimental results demonstrated the ability of the developed model in identifying and classifying faults in real time. The integration of IoT-embedded electrical drives and analytical tools provided good support for enhancing the reliability and performance of cyber–physical distributed motors. Two motor faults (BRB and ECE) were effectively modeled by imperfect corresponding representatives of the spectrum fault patterns described in the previous literature. The model also considered heterogeneity in motor sizes for simulating the impact of higher-powered motor signals on a network. Propagation of these fault patterns, with the dynamic extension of electrical and mechanical faults, stimulated interest in the diagnostic process. This simulation model facilitated the environment for testing sensor networks and data fusion approaches to facilitate better intelligence for fault identification and location. To demonstrate the spread of faulty signals within a power-line network and identify the fault type, a typical experimental IWSN motor network, based on Xbee modules and multiple-sized motors, was modeled. This architecture effectively represented a real-time industrial multi-motor network modeling environment, where all motors were connected in parallel with the main power bus. Various output results clearly compared the simulation and sensor measurements, showing close accuracy in capturing data in a real-time environment. The complexity of signals in a distributed network further complicates matters, as faulty signals can interfere with healthy ones, making it difficult to achieve high accuracy. The similarities between healthy and faulty signals lead to misclassifications, as shown in the confusion matrices. Consequently, the system’s ability to differentiate between classes is weakened, resulting in an increased error rate. Furthermore, the variability and complexity of signals from different motors exacerbate the challenge, hindering consistent diagnostic accuracy across all motors. As a result, the error rate is slightly elevated due to the intricate nature of the signals within the distributed network. Additionally, the development of the wireless node-level feature extraction technique was demonstrated for data fusion, using MCSA at the end-node-level, and decision-level fusion was implemented at the node coordinator for efficient fault diagnosis and more complex detection. 

This research establishes a strong foundation for future studies and potential areas of inquiry. However, further research is needed to explore specific aspects that require additional attention and investigation, for example:Investigate methods to further optimize cyber–physical system architecture for different industrial applications, focusing on scalability and robustness.Further simulations could improve understanding of induction motor fault frequencies and fault detection. BRB and ECE faults were considered as examples. Other faults (inter-term, bearing, stator winding) could be explored to prove signal propagation over multi-motor networks.Future expansion of distributed signature analysis and fault diagnosis could potentially utilize fuzzy logic, neuro-fuzzy, weighted fusion, D-S evidence theory, Bayesian inference, Kalman filter, and genetic algorithms.Investigate the feasibility of making the proposed framework compatible with various hardware and software platforms used in industrial IoT systems.Develop adaptive fault detection mechanisms that can automatically adjust to changing operational conditions and new types of faults in distributed motors.

## Figures and Tables

**Figure 1 sensors-24-05012-f001:**
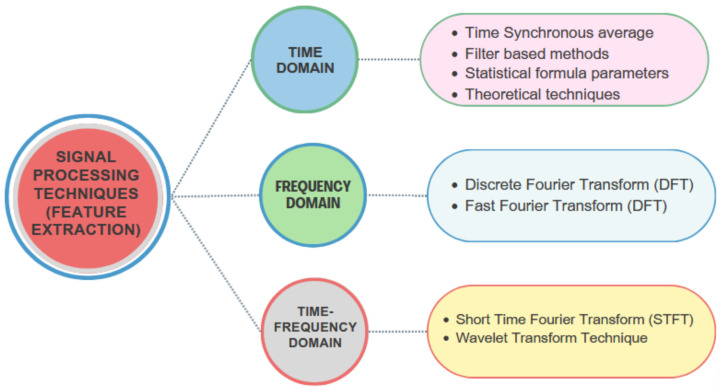
A visual representation of prevalent feature extraction methods.

**Figure 2 sensors-24-05012-f002:**
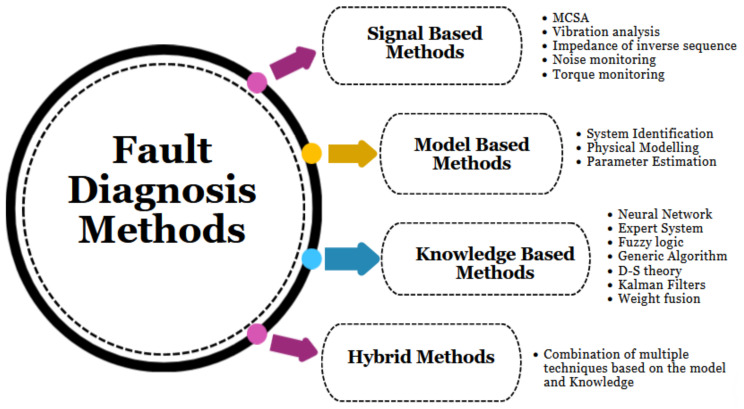
Categorizing fault diagnosis methods.

**Figure 3 sensors-24-05012-f003:**
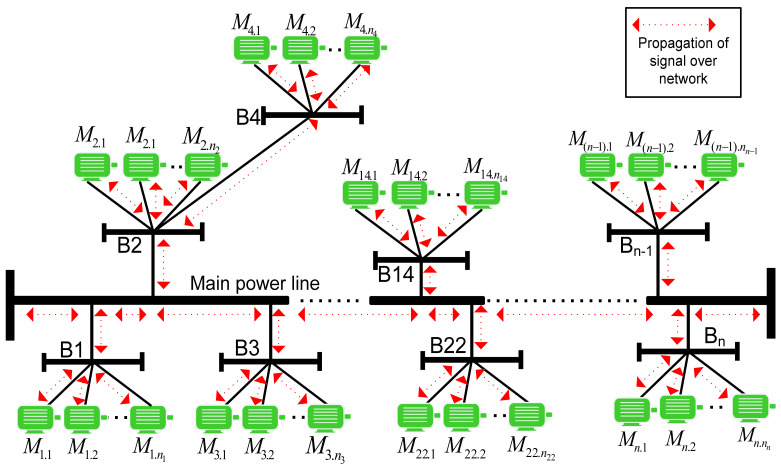
Network model for distributed industrial induction motors.

**Figure 4 sensors-24-05012-f004:**
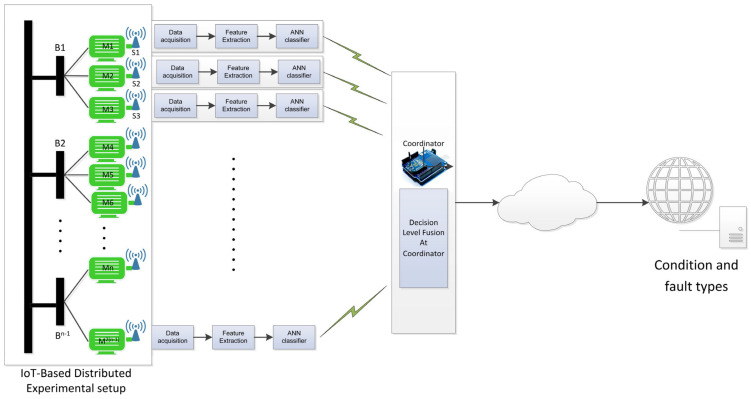
An IoT-based distributed fault diagnosis framework.

**Figure 5 sensors-24-05012-f005:**
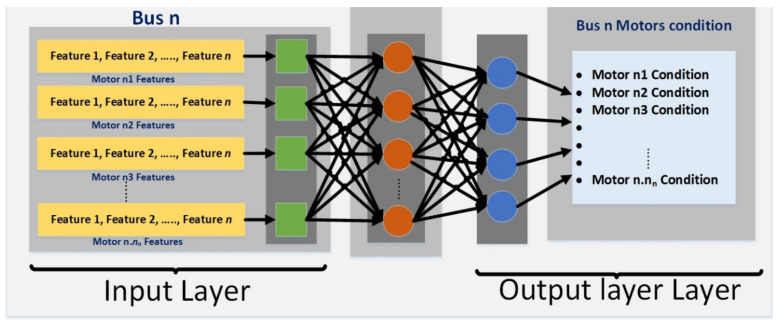
Architecture of an ANN at the distributed motor network level.

**Figure 6 sensors-24-05012-f006:**
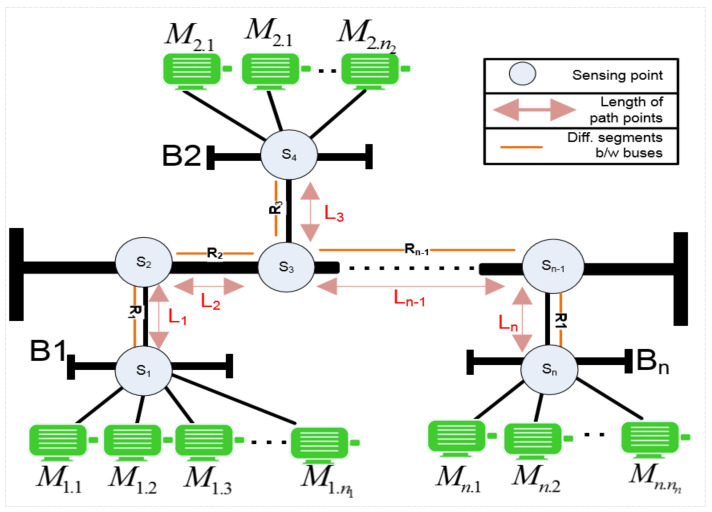
A depiction of various sensing points across a network aimed at pinpointing the positions of multiple motors.

**Figure 7 sensors-24-05012-f007:**
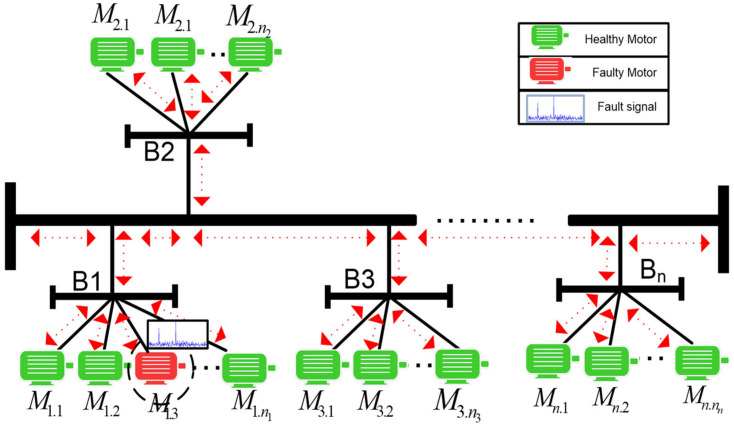
The transmission of a defective signal throughout an induction motor network.

**Figure 8 sensors-24-05012-f008:**
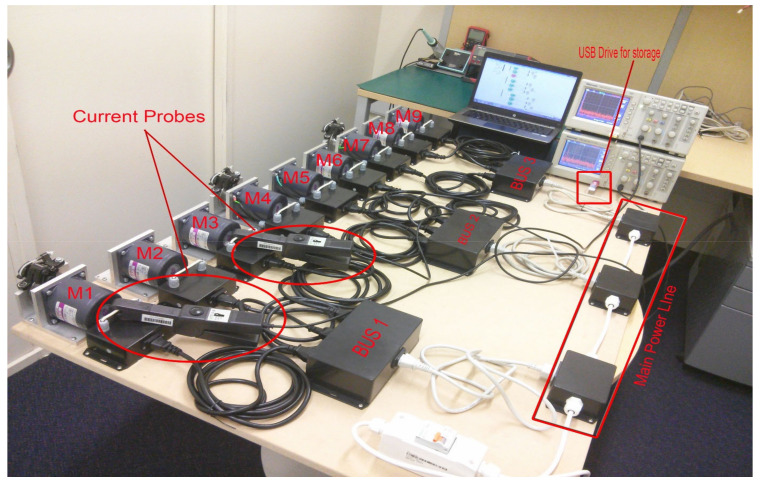
Testbed used at AUT SeNSe lab for the analysis of motor fault signatures.

**Figure 9 sensors-24-05012-f009:**
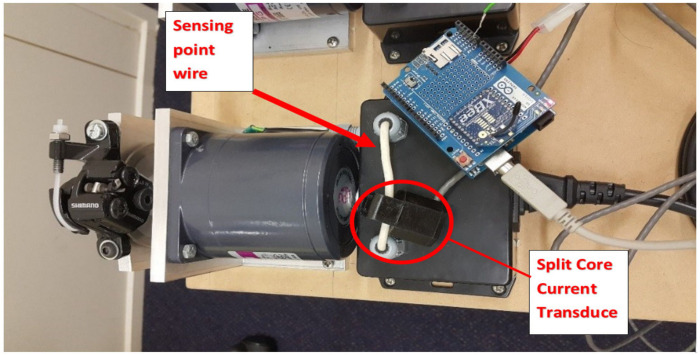
Arduino used for wireless electric current sensing and monitoring.

**Figure 10 sensors-24-05012-f010:**
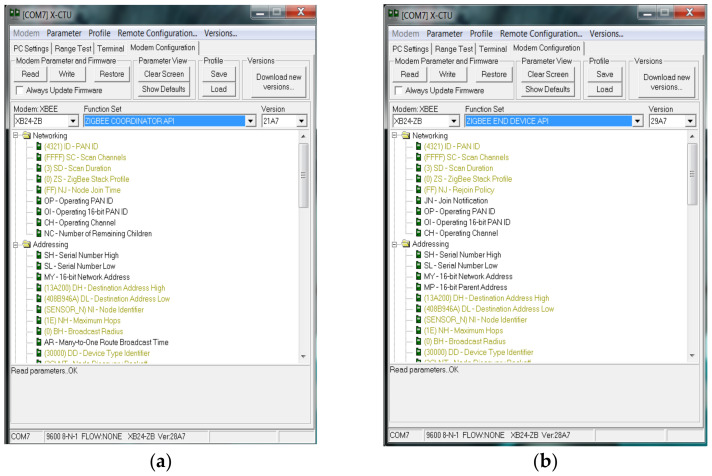
IoT-based Xbee API configuration: (**a**) coordinator; (**b**) end-node device.

**Figure 11 sensors-24-05012-f011:**
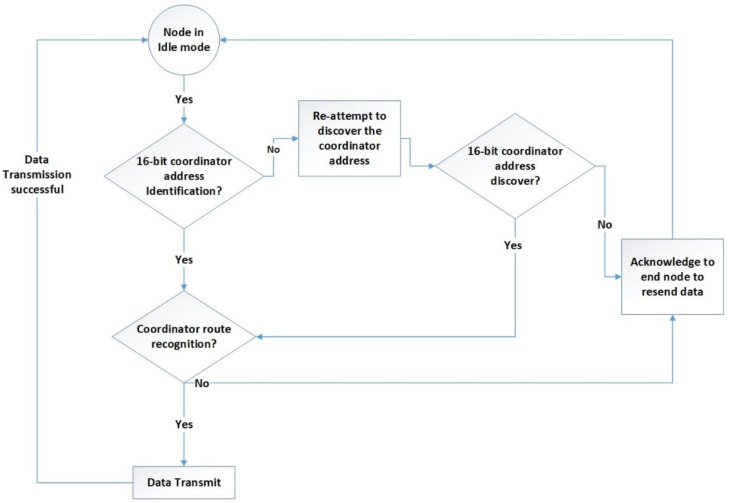
End-node transmission mode.

**Figure 12 sensors-24-05012-f012:**
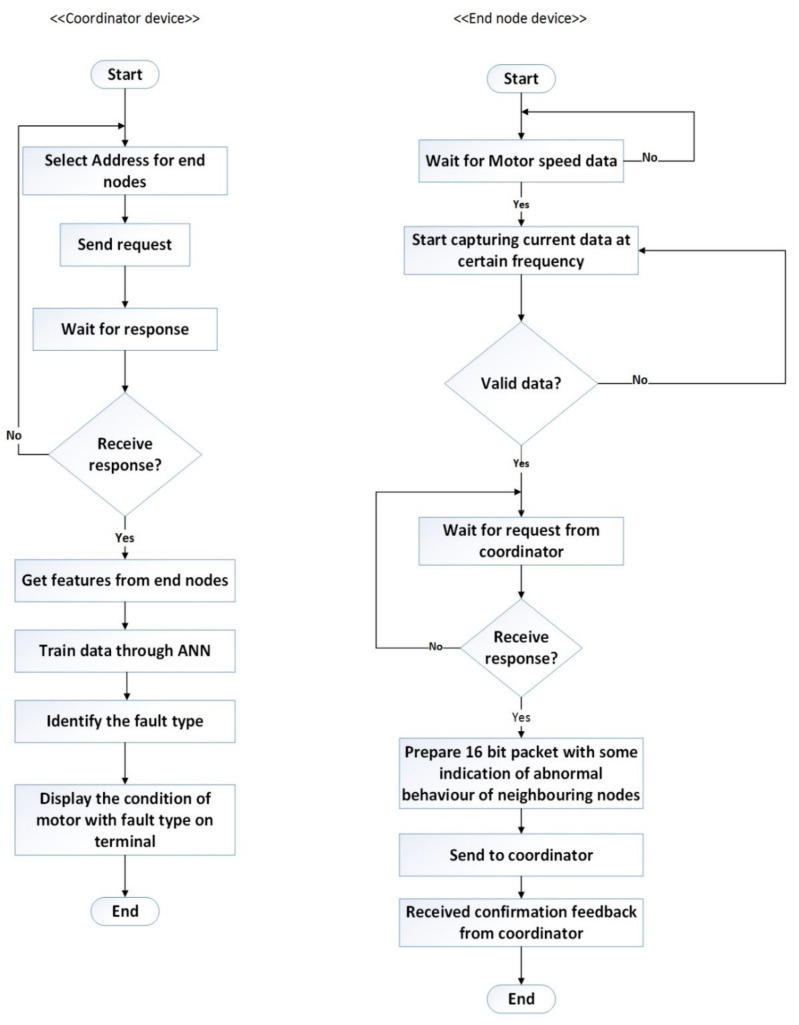
Role of coordinator and end-node devices within a network.

**Figure 13 sensors-24-05012-f013:**
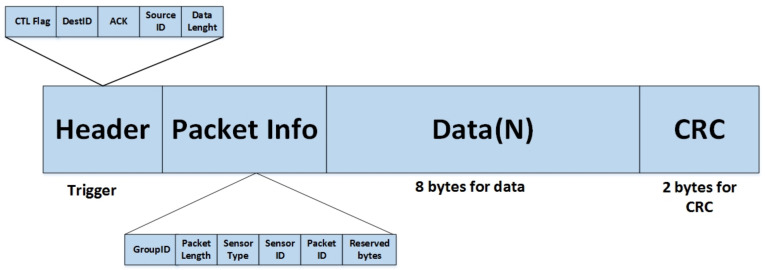
Frame structure of data packet sent to cloud.

**Figure 14 sensors-24-05012-f014:**
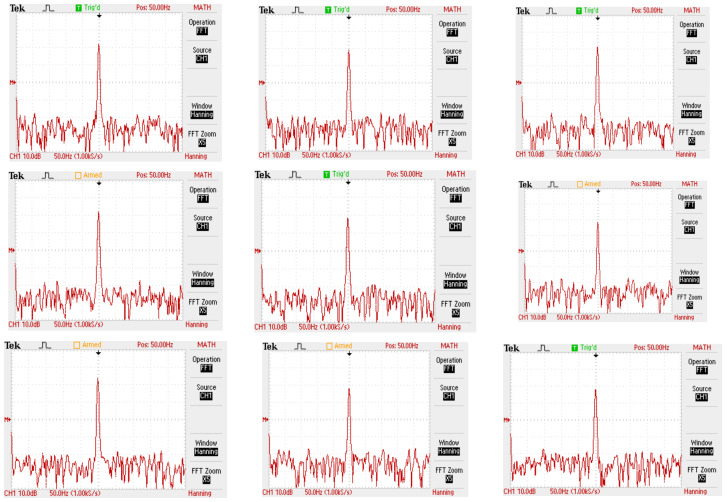
Individual FFT spectrum of electric Motor 1–9 without load.

**Figure 15 sensors-24-05012-f015:**
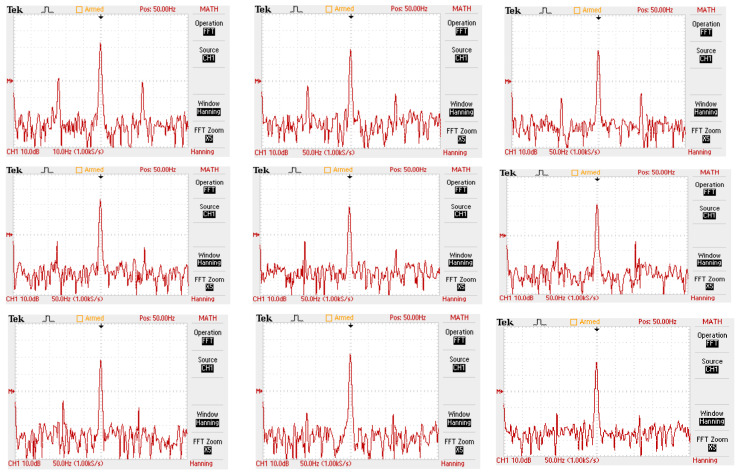
Individual FFT spectrum analysis of electric Motor 1–9 without load.

**Figure 16 sensors-24-05012-f016:**
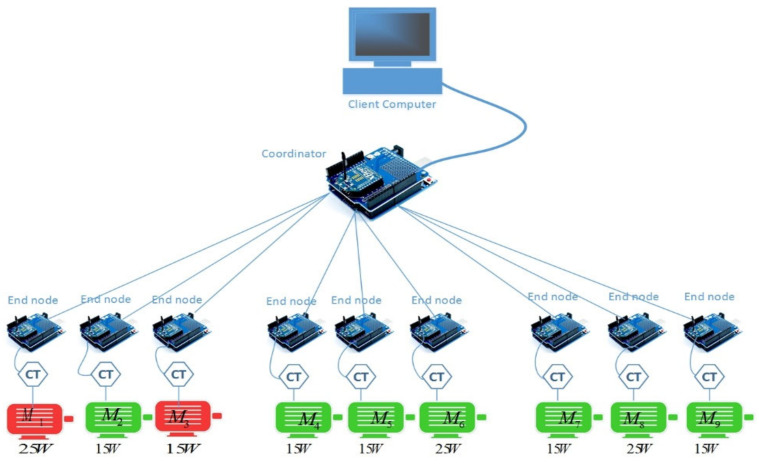
Multiple faulty motors within the same bus.

**Figure 17 sensors-24-05012-f017:**
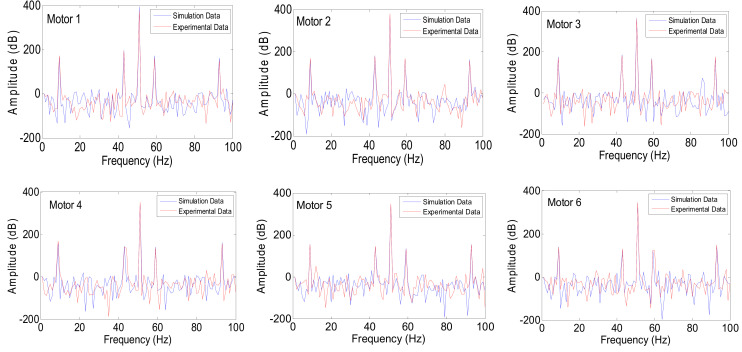
Electric current spectrum of all motors at full load in BRB and ECE fault conditions.

**Figure 18 sensors-24-05012-f018:**
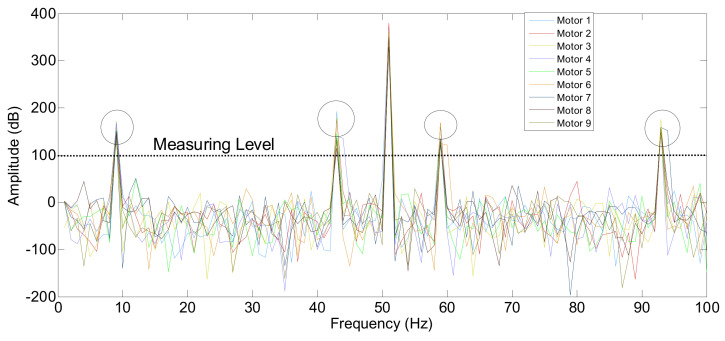
Multi-frequency fault propagation by Motor 3 and Motor 1 at full load, to observe the BRB and eccentricity fault influences on different motors.

**Figure 19 sensors-24-05012-f019:**
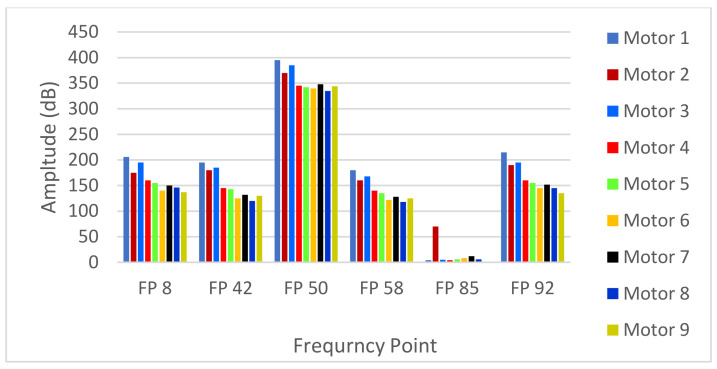
Analysis and observation chart of multi-frequency fault influence of Motor 1 and Motor 3.

**Figure 20 sensors-24-05012-f020:**
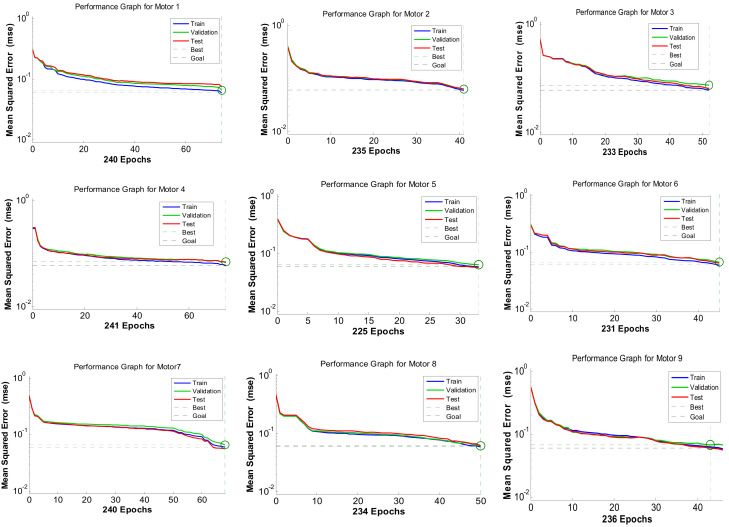
Measurement of performance graphs of all motors using neural network architecture.

**Figure 21 sensors-24-05012-f021:**
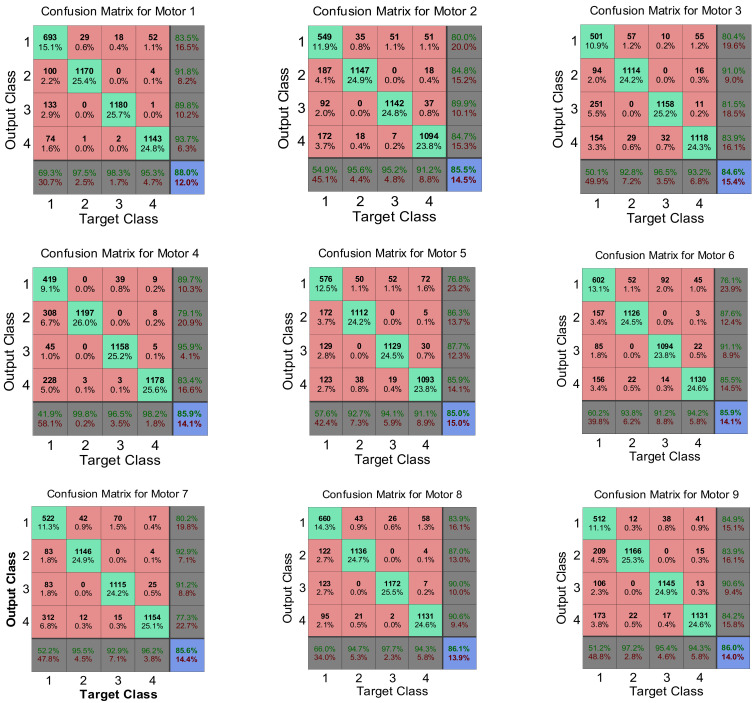
Confusion matrices of all motors using targeted and output classes.

**Figure 22 sensors-24-05012-f022:**
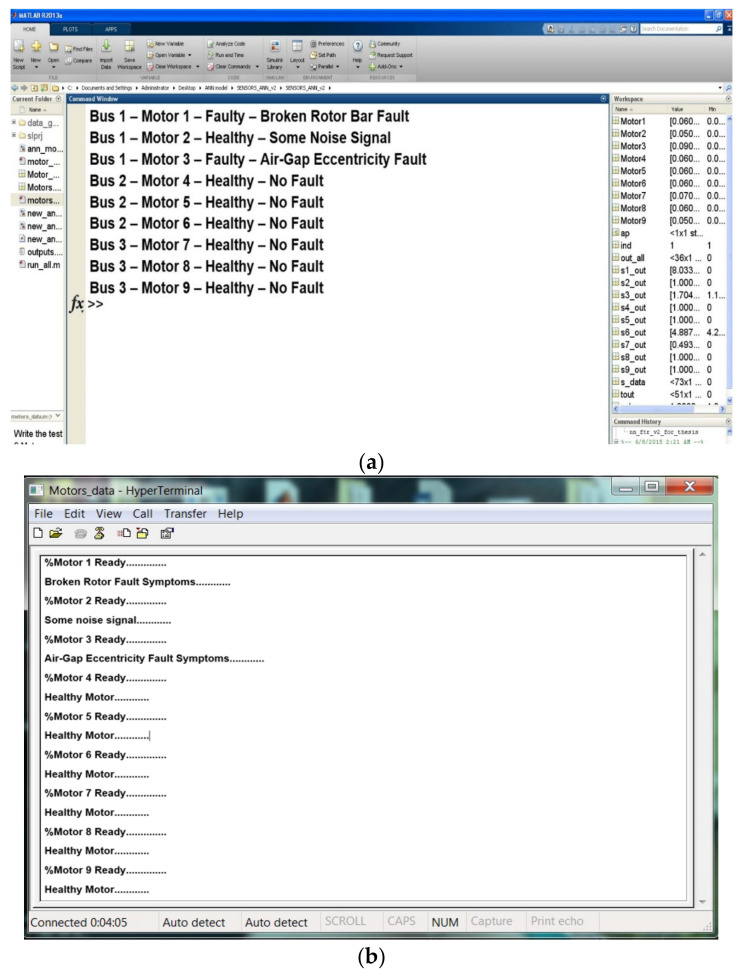
Comparison of motor condition outputs: (**a**) MATLAB simulation; (**b**) experimental results.

**Table 1 sensors-24-05012-t001:** Specifications for motors used for the testbed in the following figure in Section 5.

	Induction Motors Specifications								
	Bus1			Bus2			Bus3		
	M1	M2	M3	M4	M5	M6	M7	M8	M9
Output [w]	15 w	15 w	25 w	25 w	15 w	15 w	15 w	15 w	25 w
Current [amps]	0.26–0.33	0.26–0.33	0.54	0.54	0.26–0.33	0.26–0.33	0.26–0.33	0.26–0.33	0.54
Speed [rpm]	1200–1500	1200–1500	1250–1550	1250–1550	1200–1500	1200–1500	1200–1500	1250–1550	1200–1500
Stator winding resistance	0.6837	0.6833	0.6337	0.6820	0.6760	0.6555	0.6670	0.6837	0.6833
Rotor winding resistance	0.451	0.455	0.441	0.444	0.460	0.451	0.451	0.423	0.433
No. of poles	4	4	4	4	4	4	4	4	4
No load	1.0	1.0	1.70	1.70	1.0	0.0	1.0	1. 0	1.70
Full load	1.25	1.25	2.0	2.0	1.25	1.25	1.25	1.25	2.0

**Table 2 sensors-24-05012-t002:** Descriptions of measurement tools.

Tektronix Oscilloscopes		Current Probes Tektronix A622	
Parameter	Measuring Values	Characteristic	Value
Record length in points	1024	Frequency range	DC to 100 kHz
Sample interval	0.2466067	Max input current	100 A peak
Vertical units	dB	Output	10/100 mV/A
Horizontal units	Hz	**Standard ST-6234B Tachometer**	
Source	MATH	Range RPM	2.5 to 99,999
Operation	FFT	Max RPM resolution	0.1 RPM
Window	Hanning	Basic accuracy	+0.05% + 1d

**Table 3 sensors-24-05012-t003:** Learning data set for training process for multiple motors for case study.

Motors	Features					
	*x* _1_	*x* _2_	*x* _3_	*x* _4_	*x* _5_	*x* _6_
Motor 1	1012	0.21	[8; 42; 50; 58; 92]	[185; 195; 395; 180; 2; 160]	0.250	0.312
Motor 2	1200	0.07	[8; 42; 50; 58; 85; 92]	[165; 180; 370; 160; 60; 162]	0.147	0.200
Motor 3	1130	0.19	[8; 42; 50; 58; 92]	[175; 185; 385; 168; 5; 175]	0.142	0.191
Motor 4	1200	0.07	[8; 42; 50; 58; 92]	[160; 145; 345; 140; 4; 160]	0.231	0.290
Motor 5	1210	0.06	[8; 42;50; 58; 92]	[155; 143; 342; 135; 6; 155]	0.122	0.210
Motor 6	1280	0.01	[8; 42; 50; 58; 92]	[140; 125; 340; 122; 8; 145]	0.150	0.215
Motor 7	1250	0.03	[8; 42; 50; 58; 92]	[150; 132; 348; 128; 12; 152]	0.164	0.210
Motor 8	1280	0.01	[8; 42; 50; 58; 92]	[146; 120; 335; 118; 6; 145]	0.150	0.190
Motor 9	1250	0.03	[8; 42; 50; 58; 92]	[137; 130; 344; 125; 1; 135]	0.164	0.210

**Table 4 sensors-24-05012-t004:** Best performances for classification for case study.

Motors	MSE Performance	No. of Epochs	Accuracy (%)
$M_{1}$	0.0596	240	88.0
$M_{2}$	0.0631	235	85.5
$M_{3}$	0.0617	233	84.6
$M_{4}$	0.0622	241	85.9
$M_{5}$	0.0613	225	85.0
$M_{6}$	0.0622	231	85.9
$M_{7}$	0.0610	240	85.6
$M_{8}$	0.0600	234	86.1
$M_{9}$	0.0629	236	86.0

## Data Availability

The datasets generated and/or analyzed during the current research are available from the corresponding author on reasonable request.

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
