# Peer review of "Cyber–Physical Distributed Intelligent Motor Fault Detection"

_sensors, 2024, doi:10.3390/s24155012_

Round 1

Reviewer 1 Report

Comments and Suggestions for Authors

The paper explores the problem of fault detection in distributed motors from the perspective of the Internet of Electrical Drives (IoED), developing a cyber-physical system architecture benefiting from the distributed Internet of Things (DioT) idea and utilizing Fast Fourier Transforms (FFT) and Artificial Neural Networks (ANN) to improve the accuracy and sensitivity of fault detection. The experimental part is well done and most of the article is well written, but there are still some areas that need to be improved:

1. In the second part of Related Works, the figure serial number should be 1.2 there is an error in the serial number of the figure labeling, and the figure serial number in line 140.149.182.184 is wrong.

2. I think in the signal processing part, we can try to use various methods for comparison, such as wavelet transform (WT) , Synchrosqueezed wavelet transforms (SWT), short-time Fourier transform (STFT), etc., which may improve the accuracy and sensitivity of fault detection.

3. The experimental part is well done, but I think there is a lack of some comparative experiments, the article needs to be compared with some other methods to reflect the advantages of this paper.

Author Response

Reviewer Comment: In the second part of Related Works, the figure serial number should be 1.2 there is an error in the serial number of the figure label, and the figure serial number in line 140.149.182.184 is wrong.

Authors Response: Thank you for your suggestion. The errors have been rectified and modified as per your recommendations.   

Edit to Manuscript:  All figure numbers have been updated according to the recommendations. Figure 1 and Figure 2 have been corrected as per your suggestion. These changes are now reflected in lines 150, 152, 192, and 194.

Reviewer Comment:

  • I think in the signal processing part, we can try to use various methods for comparison, such as wavelet transform (WT), Synchro squeezed wavelet transforms (SWT), short-time Fourier transform (STFT), etc., which may improve the accuracy and sensitivity of fault detection.
  • The experimental part is well done, but I think there is a lack of some comparative experiments, the article needs to be compared with some other methods to reflect the advantages of this paper.

Authors Response: Thank you for your valuable suggestion. We plan to include a comparison of these techniques in our upcoming paper and have highlighted this aspect in the future directions section of this paper. As suggested, we have removed the discussion of these techniques on page 4 since they are not directly relevant to the paper's goals.

Edit to Manuscript:  The future recommendations section at the end of the conclusion has been revised as suggested, reflected in lines 708-710

Reviewer 2 Report

Comments and Suggestions for Authors

The authors presented a cyber-physical distributed intelligent motors fault detection method and then an experiment was performed to validate it. It is meaningful and some revision could be performed as below.

1. This paper title can be revised as cyber-physical distributed intelligent motors fault detection.

2. Figure 21 shows the confusion matrices of all motors using targeted and output classes, but it is seen that the diagnosis accuracy is very low.

3. Figure 17 shows the electric current spectrum of all motors at full-load in BRB and ECE fault conditions, and it is questionable that a digital twin or numerical simulation is used to generate the simulated samples trained?

4. Some references could be considered for fault detection in signal processing, such as coupled neurons with multi-objective optimization benefit incipient fault identification of machinery; harmonic-Gaussian double-well potential stochastic resonance with its application to enhance weak fault characteristics of machinery.

Author Response

Reviewer Comment: This paper title can be revised as cyber-physical distributed intelligent motors fault detection.

Authors Response: Thank you for your suggestion. The title of the paper has been updated according to the suggestion.

Edit to Manuscript:  Line 2: The title of the paper has been updated according to the suggestion.

Reviewer Comment: Figure 21 shows the confusion matrices of all motors using targeted and output classes, but it is seen that the diagnosis accuracy is very low.

Authors Response: The overlapping characteristics of healthy and faulty signals lead to misclassification, as seen in the confusion matrices. The system's ability to distinguish between different classes is compromised due to these interfering signals, resulting in a higher error rate. Additionally, the complexity and variability of the signals from different motors add to the difficulty, making it hard to maintain consistent diagnosis accuracy across all motors. Therefore, the error rate is slightly higher due to the intricate nature of the signals within the distributed network.

Edit to Manuscript:  The justification has been added in the manuscript at lines 643 to 648.

Reviewer Comment: Figure 17 shows the electric current spectrum of all motors at full load in BRB and ECE fault conditions, and it is questionable that a digital twin or numerical simulation is used to generate the simulated samples trained?

Authors Response: We developed a physical testbed and created a simulink model to validate the concept. To achieve this, we first acquired data from the motors using CT sensors and stored the data in a storage device. Then, we simulated the behaviour using the Simulink model and acquired the signals. We also acquired signals from an oscilloscope to ensure accuracy and reliability. We analyzed the oscilloscope data alongside the simulated data to verify the consistency and correctness of the signals. After this validation process, we trained the data using adopted methodologies. This approach allowed us to ensure that the simulated samples used for training were accurate representations of the actual motor conditions under BRB and ECE fault conditions.

Reviewer 3 Report

Comments and Suggestions for Authors

The subject of the study of this paper was the process of fault detection in distributed motors through the use of the Internet of Electrical Drives (IoED) trend in the aspect of cyber-physical intelligence for solving a scientific problem.

The authors carried out their technical approach/methods based on comprehensive research inquiries both through an in-depth critical analysis of the literature on the subject of the research, an in-depth analysis, a designed mathematical model and carried out research and simulation tests (analysis, mathematical model of the cyber-physical system, simulations, etc.). ), as well as, most importantly, by verifying the results obtained in terms of the cyber-physical approach for the detection of distributed motors failures through the use of a practical experiment, such as a case study, and on this basis, citing relevant insights and formulating final conclusions reflected in practical applications. 

This presented manuscript, developed at a very high research level in terms of content and slightly weaker in the field of methodology, is characterized by a very high research level, making an important scientific contribution to the advancement of knowledge in various fields, including electrical engineering, electronics, or electrical power engineering, within the broad area of electrical machinery, and mainly the area of knowledge including technical diagnostics, reliability analysis, and selected probability elements for the development of a process on cyber-physical distributed motors fault detection intelligence.

In terms of research, the work is characterized by an interesting technical approach in solving the research problem. Its new and useful contribution to the presentation of substantive knowledge to the potential reader is the technical approach itself, the apt methods used (IoED, DIoT, FFT, ANN, etc.) and its in-depth analysis ( figures, diagrams, structures, matrices, tables, etc.) in the field of the use of interdisciplinary knowledge, including mainly basing the research problem on the targeting of the designed system based on the use of cyber-physical intelligence of distributed motors fault detection.  

For example, to confirm the above assessment, it should be emphasized that the authors of this manuscript not only presented substantive knowledge in accordance with the subject matter of the research, discussed and analysed in detail their research inquiries to solve the scientific problem, but most importantly, with the help of the integration of IoT (Internet of Things) embedded electric drives, IoED concepts with applied analytical tools for enhancing the reliability and performance of distributed cyber-physical motors, including experimental validation of the results obtained based on the practical experiment conducted for the selected experimental procedures, modeled the IWSN (Industrial Wireless Sensor Network) motor network for the design cyber-physical distributed motors fault detection intelligence, etc.). 

Moreover, the authors of this paper paid special attention to the development process of the feature extraction technique at the wireless node level, performing an in-depth analysis of the current signatures of the motor under study MCSA (Motor Current Signature Analysis) to achieve efficient fault diagnosis and more advanced detection, and thus set a development trend in future fault localization research for network nodes.

In terms of methodology, including the layout and structure of the construction of the presented work, in addition to the introduction in accordance with the subject of the study (Chapter 1), based on the critical analysis of the literature made, the authors presented related work on various fault diagnosis techniques for electrical machines (Chapter 2) in the field of signature analysis of distributed motor networks and diagnosis of selected types of faults (Chapter 3) for the purpose of cyber-physical intelligence for fault detection of distributed motors, which are the subject of consideration in this work. 

Subsequent chapters of this work (Chapters 4-6), the authors devoted both the process of network mathematical modeling of distributed motors (Chapter 4) and, above all, the verification of the results obtained through the design of an experimental multi-motors test-bed and results obtained (Chapter 5), for the purpose of discussing and validating the results obtained through the application of analytical research inquiries, which allowed them in the final part of the work to cite important insights and formulate conclusions reflected in practical applications (Chapter 6). 

Additionally, in the aspect of confirming the evaluation of the reviewed manuscript, other examples can be cited, among others, it should be emphasized that the authors of this article, in addition to the professional graphical presentation (Figures 1-22), including in the form of block diagrams in relation to the analytical analysis (Figures 1-7), and Figures 8-13, 16 and 21-22 (block diagrams), as well as the simulations performed (Figures 14-15 and 17-20), in the field of the experimental verification of the results obtained, conducted an in-depth analysis of the proposed solution. 

It should be noted that the proposed strategy for the problems related to cyber-physical distributed motors fault detection intelligence was done not only through the presented analytical studies supported by the applied methods (IoED, DIoT, ANN, etc.) using Fast Fourier Transform (FFT), or Short-Time Fourier Transform (STFT), as well as wavelet transform through the cited layouts, schematics, etc. (Figs. 1-7) and tabular summaries of key parameters (Tables 1-4). 

But mainly through the proposed approach (Figs. 8-12) in the form of the design of cyber-physical distributed motors fault detection intelligence with simulation tests performed, e.g., Figs. 15, 17-19 (analytical studies), including mainly through the process of validation of the results obtained in accordance with the developed model of the proposed system (Fig. 23) and the simulations performed (Figs. 14-15 and 17-20) in the field of modeling the proposed system, and in terms of experimental verification using the designed practical experiment, for the purpose of verifying and validating the obtained results in terms of the scientific problem that concerns them, which allowed them to formulate important insights and final conclusions that are reflected in practical applications. 

During the review of this paper, except for minor editing errors and some inaccuracies of methodological nature, no other shortcomings were observed in this manuscript, having a key impact on the level and quality of the work presented. 

Abstract:

In accordance with the recommendations of reputable publishing houses and journals, e.g., IEEE TTE, IEEE Access, Wiley and Sons, or MDPI, part of the abstract should contain the following basic elements: introduction (reference to the subject matter of the study), clear definition of the aim of the work, approximation/addressing of the potential solution to the problem/methods, and response to on the basis of the research, test, experience, developed mathematical model, to formulate relevant observations and final conclusions. 

The abstract should not exceed 200 words, in this manuscript there are 206 of them. In my opinion, the abstract, apart from failing to clearly state the purpose of the work presented, did not explicitly address the important insights and conclusions made (part of the conclusion of the paper).

 Minor inaccuracies regarding the abstract of this article: 

1. There is no explicit aim for this article. 

2. It is rather not advisable to explain abbreviations and designations in the abstract, in the case of abbreviations, e.g. IoED, points 12 and 25, respectively; DIoT, point 14; FFT, point 15, or ANN, point 16. It should be noted that these are not serious shortcomings, but in my opinion, the abstract should focus on other important elements (purpose, results achieved, methods used, conclusions, etc.). In my opinion, the abbreviations and designations used in the abstract should be explained in the rest of the paper, i.e., for example, in the introduction, key words, appendix, or in the rest of the paper. 

3. Lack of explicit reference to both the expected and predicted results of the research (analysis, model, simulation, experiment) and reference to the relevant insights and final conclusions reflected in practical applications. 

The rest of the work:

1. In the case of the presentation of the literature references of the subject of the study, I observed in the paper both a failure to follow the order of writing and multiple duplication of references, including references in the form of large blocks. For example, [1-7], p. 1, points 32 and 45, respectively; [8] and [9], p. 1, points 51 and 85; [9], p. 1, points 60 and 95, or within the scope of [14], p. 6, points 251, points 260 and 264, etc., respectively. This is all the more incomprehensible since the authors, in their paper of 26 pages, presented only 14 items of literature on the subject of the study. Please review the entire paper in this regard and make appropriate corrections.

2. In this manuscript, I observed multiple explanations of some abbreviations both in the abstract and in the rest of the paper. For example, ANN on page 5, points 211-212; page 6, points 215 and 227, respectively, or FFT on page 17, point 529, etc. Please review the entire manuscript in this regard and make the appropriate corrections.

3. No explanation of some abbreviations cited in this paper, e.g. UART, ECE, etc. Please review the entire article in this regard.

4. Inaccuracies in the presentation of some figures and tables, especially in the order of numbering, and thus references to these figures, which is a serious oversight by the authors of this paper. For example, on page 11, the authors refer to Table 1, which I did not find in this paper, similarly: the first figure placed in the paper, is Figure 19 on page 4, etc., etc... ...Please explain this approach, as it is completely incomprehensible to me. I believe that Sensors magazine is a serious MDPI publication, and one should check the article in detail both in terms of content and in terms of methodology before submitting it.

5. In this paper, the authors frequently cite the methods used, including FFT, DFT, STFT, PDF, or wavelet transform, while I did not find a single formula in the paper. Similarly, in the case of the modeling process, the authors based their project on the analysis and presentation of structural models for the development of a cyber-physical system architecture for effective fault detection, using an approach based on neural networks. Please explain this type of approach for solving a scientific problem.

6.There are no references in the text of this paper to some figures and tables, e.g. Figures 1,4,7,11-13, 22, as well as tables (Tables 3 and 5-6). Furthermore, the numbering of the table presented on page 25 is incorrect. Please review the entire work in this regard and make the appropriate correction.

7. In this paper, I observed incorrect references to some figures or tables. For example, on page 3, the authors refer to Figure 10 and present Figure 19 on page 4; on page 5, they refer to Figure 3 and present Figure 3, noting it is Figure 2 presented in this manuscript. Then in Chapter 3, points 280-281, the authors refer again to Figure 3 and present the wrong numbering of the figure on page 7, point 283, etc., etc... In conclusion, the numbering of the posted figures is incorrect, hence it is difficult to count the number of drawings without renumbering. Please review the entire work in this regard and make a proper arrangement of figures and tables, making the appropriate correction.

8.Referring to point 8, for example, in the manuscript on page 17 the authors refer to Fig. 5.3, point 541, and present Fig. 14, or on page 16 to Fig. 5.1, point 522. In addition, I observed duplicate references to some drawings, such as Fig. 4 on page 8, or Fig. 6 on page 10. Please review the entire work in this regard and make the appropriate correction.

9. In methodological terms, in my opinion, it is not advisable to present figures/tables or formulas in the concluding section of a chapter/subchapter, etc., as was done in this article in the case of Figure 5 on page 9 in Chapter 3; Figure 7 on page 11 in Chapter 4, or Figure 22 on page 23 in Chapter 4. Please make a correction by reviewing the entire manuscript in this regard.

10.  Sentence sequences that are too short, such as page 15, point 511, or page 16, points 522-523.

11. The final conclusions should be supported by the research results obtained, and in the case of an article of a review nature, by the results obtained based on the in-depth analysis of the issue under consideration. This is all the more incomprehensible in view of the fact that the reviewed manuscript was developed by the authors at a very high level of merit, especially in terms of research inquiries devoted to cyber-physical distributed motors fault detection intelligence. In addition, the research performed in the field of the in-depth analysis carried out and the results obtained were discussed through the requirements according to the subject of the research, graphical depiction (Figures 1-22), especially in terms of simulation studies (Figures 14-15 and 17-20), including structural models (Figures 3-7 and 21) and tabulated (Table 1-4), and verified through the developed practical experiment, among others, using case studies.

12. Minor errors in the literature list appended at the end of the paper regarding editing and use of punctuation marks, e.g. p. 25, items 8, 10 and items 13-14. 

Strong aspects:

The technical approach/methods used, the idea of solving the problem and its explanation, the analysis of the research results obtained, supported by structural models, the methods used, simulations, the practical experiment developed, etc., and the formulation of the final conclusions, the relevance in terms of the methods used and the ability to use them.

Weak aspects:

Many inaccuracies, mainly in the methodological aspect, which do not significantly affect the quality of the reviewed work, i.e. editorial errors and poor quality of the methodological part of the abstract, however, require detailed correction.

Recommended changes:

Regardless of the Editor's decision, at this stage of the paper, I would recommend that the authors improve the paper according to the recommendations in points 1-12 of this review and the above-mentioned point (weak aspects). 

Author Response

Reviewer Comment: There is no explicit aim for this article.

Authors Response and Edit to Manuscript: Thank you for your comments. The impacts and importance of the topic in the industry have been thoroughly explained in the introduction section. The abstract has been revised to clearly emphasize the paper's aim.

Reviewer Comment: It is rather not advisable to explain abbreviations and designations in the abstract, in the case of abbreviations, e.g. IoED, points 12 and 25, respectively; DIoT, point 14; FFT, point 15, or ANN, point 16. It should be noted that these are not serious shortcomings, but in my opinion, the abstract should focus on other important elements (purpose, results achieved, methods used, conclusions, etc.). In my opinion, the abbreviations and designations used in the abstract should be explained in the rest of the paper, i.e., for example, in the introduction, key words, appendix, or in the rest of the paper.

Authors Response: The required suggestion has been incorporated, and the abbreviations have been removed from the abstract.

Edit to Manuscript: The abbreviations from the abstract have been removed and defined at the end of the paper in a list of abbreviations.

Reviewer Comment: Lack of explicit reference to both the expected and predicted results of the research (analysis, model, simulation, experiment) and reference to the relevant insights and final conclusions reflected in practical applications.

Authors Response and Edit to Manuscript: All related references have been added and properly cited in the text to support the concept. The mistakes in Figures 1 and 2, which caused confusion in judgmental calls and analytical referencing, have been reviewed and corrected. Additionally, the abstract and conclusion have been amended to clarify the referencing flow of the paper.

Reviewer Comment: In the case of the presentation of the literature references of the subject of the study, I observed in the paper both a failure to follow the order of writing and multiple duplication of references, including references in the form of large blocks. For example, [1-7], p. 1, points 32 and 45, respectively; [8] and [9], p. 1, points 51 and 85; [9], p. 1, points 60 and 95, or within the scope of [14], p. 6, points 251, points 260 and 264, etc., respectively. This is all the more incomprehensible since the authors, in their paper of 26 pages, presented only 14 items of literature on the subject of the study. Please review the entire paper in this regard and make appropriate corrections

Authors Response: Thank you for the valuable comment. Some of the references are review papers that each contain several studies. We have incorporated your suggestions and made the necessary revisions to the paper.

Edit to Manuscript: References have been modified as per the suggestion, and additional references have been provided to enhance the literature review in the related works section. References [15] to [26] have been added to the manuscript to strengthen the paper's foundation.

Reviewer Comment: In this manuscript, I observed multiple explanations of some abbreviations both in the abstract and in the rest of the paper. For example, ANN on page 5, points 211-212; page 6, points 215 and 227, respectively, or FFT on page 17, point 529, etc. Please review the entire manuscript in this regard and make the appropriate corrections.

Authors Response: The required suggestion has been incorporated.

Edit to Manuscript: All repetitive abbreviations have been carefully rectified, and the duplication has been removed.

Reviewer Comment: No explanation of some abbreviations cited in this paper, e.g. UART, ECE, etc. Please review the entire article in this regard.

Authors Response: The explanation of UART has already been provided in lines 476 to 482, detailing its function and significance in serial communication. Additionally, the configuration steps for UART are illustrated in Figure 10, which visually guides the setup process and highlights key parameters for optimal performance. The abbreviation and further explanations added in the paper to clarify the concept.

Reviewer Comment: Inaccuracies in the presentation of some figures and tables, especially in the order of numbering, and thus references to these figures, which is a serious oversight by the authors of this paper. For example, on page 11, the authors refer to Table 1, which I did not find in this paper, similarly: the first figure placed in the paper, is Figure 19 on page 4, etc., etc... ...Please explain this approach, as it is completely incomprehensible to me. I believe that Sensors magazine is a serious MDPI publication, and one should check the article in detail both in terms of content and in terms of methodology before submitting it.

Authors Response: Thank you for your comments regarding the figures and tables. We have carefully reviewed the presentation and corrected any inaccuracies in the numbering and figures references. This ensures that the information is now clearly organized and accurately reflects the content of the manuscript. We appreciate your attention to detail, which has helped enhance the overall clarity of our work.

Edit to Manuscript: Corrected all inaccuracies in the numbering and figures references.

Reviewer Comment: In this paper, the authors frequently cite the methods used, including FFT, DFT, STFT, PDF, or wavelet transform, while I did not find a single formula in the paper. Similarly, in the case of the modeling process, the authors based their project on the analysis and presentation of structural models for the development of a cyber-physical system architecture for effective fault detection, using an approach based on neural networks. Please explain this type of approach for solving a scientific problem.

Authors Response: Because we used the traditional FFT algorithm and mathematical equations, we did not discuss them in detail. The focus of the paper is on formulation and proving the concept, and we have included relevant mathematical components and diagrams to enhance understanding for both the reviewer and the audience. Additionally, we employed a neural network-based approach to model the fault type recognition. This enables efficient fault detection in distributed motor environments. This approach leverages the network's ability to learn complex patterns and relationships within the data, ultimately enhancing reliability and performance in fault identification. By integrating these methodologies, our work aims to advance the field of fault detection and contribute to more robust industrial applications.

Edit to Manuscript: References to wavelet and other transforms, except for FFT, have been removed from the paper as they have minor association with the paper's goals.

Reviewer Comment: There are no references in the text of this paper to some figures and tables, e.g. Figures 1,4,7,11-13, 22, as well as tables (Tables 3 and 5-6). Furthermore, the numbering of the table presented on page 25 is incorrect. Please review the entire work in this regard and make the appropriate correction.

Authors Response and Manuscript Edit: All figure numbers and table citations in the text have been carefully checked for accuracy. We ensured that all references to figures and tables are correctly aligned with their respective placements in the manuscript to maintain clarity and coherence throughout the document. Figures 1 and 2 were initially assigned incorrect numbers, which have now been corrected. Additionally, the table numbering, which began at 2, has been adjusted throughout the document.

Reviewer Comment: In this paper, I observed incorrect references to some figures or tables. For example, on page 3, the authors refer to Figure 10 and present Figure 19 on page 4; on page 5, they refer to Figure 3 and present Figure 3, noting it is Figure 2 presented in this manuscript. Then in Chapter 3, points 280-281, the authors refer again to Figure 3 and present the wrong numbering of the figure on page 7, point 283, etc., etc... In conclusion, the numbering of the posted figures is incorrect, hence it is difficult to count the number of drawings without renumbering. Please review the entire work in this regard and make a proper arrangement of figures and tables, making the appropriate correction.

Referring to point 8, for example, in the manuscript on page 17 the authors refer to Fig. 5.3, point 541, and present Fig. 14, or on page 16 to Fig. 5.1, point 522. In addition, I observed duplicate references to some drawings, such as Fig. 4 on page 8, or Fig. 6 on page 10. Please review the entire work in this regard and make the appropriate correction.

Authors Response and Edit to Manuscript: All figure numbers and table citations in the text have been thoroughly checked for accuracy. Figures 1 and 2 have been corrected as per your recommendation, with changes reflected in lines 150,152,192 and 194. Additionally, the issue with table numbering, which incorrectly started from Table 2, has been resolved. The authors have carefully reviewed the entire manuscript to ensure that all figures and tables are accurately cited and discussed in the text

Reviewer Comment: In methodological terms, in my opinion, it is not advisable to present figures/tables or formulas in the concluding section of a chapter/subchapter, etc., as was done in this article in the case of Figure 5 on page 9 in Chapter 3; Figure 7 on page 11 in Chapter 4, or Figure 22 on page 23 in Chapter 4. Please make a correction by reviewing the entire manuscript in this regard.

Authors Response: All sections of the paper have been thoroughly proofread to eliminate any errors and improve clarity. This meticulous review process has enhanced the overall quality of the manuscript, ensuring that the content is accurate, coherent, and well-organized. We believe that these improvements contribute to a more professional presentation of our research findings.

Reviewer Comment: Sentence sequences that are too short, such as page 15, point 511, or page 16, points 522-523.

Authors Response: The required suggestion has been incorporated.

Edit to Manuscript: Lines 535 to 538 and lines 637 to 643 have been rephrased, resulting in lines 539 to 543 and lines 643 to 648, which are now of reasonable length.

Reviewer Comment: The final conclusions should be supported by the research results obtained, and in the case of an article of a review nature, by the results obtained based on the in-depth analysis of the issue under consideration. This is all the more incomprehensible in view of the fact that the reviewed manuscript was developed by the authors at a very high level of merit, especially in terms of research inquiries devoted to cyber-physical distributed motors’ fault detection intelligence. In addition, the research performed in the field of the in-depth analysis carried out and the results obtained were discussed through the requirements according to the subject of the research, graphical depiction (Figures 1-22), especially in terms of simulation studies (Figures 14-15 and 17-20), including structural models (Figures 3-7 and 21) and tabulated (Table 1-4), and verified through the developed practical experiment, among others, using case studies.

Authors Response: Thank you for your thoughtful comments and recognition of the merit of our manuscript. We appreciate your emphasis on the importance of supporting our conclusions with research findings. In response, we have ensured that our final conclusions are now explicitly tied to the results obtained from our in-depth analysis, as well as the data presented in the figures and tables throughout the manuscript.

Edit to Manuscript: We have highlighted key findings from the simulation studies and case studies to reinforce the validity of our conclusions. (lines 697 to714)

Round 2

Reviewer 1 Report

Comments and Suggestions for Authors

1. The author has rectified an error in the numbering of the image labeling in the second part of Related Works, wherein the image number should be 1.2, and an error in the numbering of the image in line 140.149.182.184. The revised manuscript is free of these issues.

2. In response to the second question, the authors have made the suggested revisions in the "Future Recommendations" section, which can be found at the end of the conclusions.

I am now in a position to recommend this paper for publication.

Author Response

Reviewer Comment: The author has rectified an error in the numbering of the image labeling in the second part of Related Works, wherein the image number should be 1.2, and an error in the numbering of the image in line 140.149.182.184. The revised manuscript is free of these issues.

Authors Response: Thank you for reviewing our manuscript and accept modifications.

Reviewer Comment: In response to the second question, the authors have made the suggested revisions in the "Future Recommendations" section, which can be found at the end of the conclusions.

Authors Response: Thank you for reviewing our manuscript and accept modifications.